# Metagenomic Strain-Typing Combined with Isolate Sequencing Provides Increased Resolution of the Genetic Diversity of *Campylobacter jejuni* Carriage in Wild Birds

**DOI:** 10.3390/microorganisms11010121

**Published:** 2023-01-03

**Authors:** Malte Herold, Louise Hock, Christian Penny, Cécile Walczak, Fatu Djabi, Henry-Michel Cauchie, Catherine Ragimbeau

**Affiliations:** 1Environmental Research and Innovation (ERIN) Department, Luxembourg Institute of Science and Technology (LIST), 41 rue du Brill, L-4422 Belvaux, Luxembourg; 2Epidemiology and Microbial Genomics, Laboratoire National de Santé (LNS), 1 rue Louis Rech, L-3555 Dudelange, Luxembourg

**Keywords:** *Campylobacter jejuni*, strain typing, metagenomics, wild birds, molecular epidemiology

## Abstract

As the world’s leading cause of human gastro-enteritis, the food- and waterborne pathogen *Campylobacter* needs to be intensively monitored through a One Health approach. Particularly, wild birds have been hypothesized to contribute to the spread of human clinical recurring *C. jejuni* genotypes across several countries. A major concern in studying epidemiological dynamics is resolving the large genomic diversity of strains circulating in the environment and various reservoirs, challenging to achieve with isolation techniques. Here, we applied a passive-filtration method to obtain isolates and in parallel recovered genotypes from metagenomic sequencing data from associated filter sweeps. For genotyping mixed strains, a reference-based computational workflow to predict allelic profiles of nine extended-MLST loci was utilized. We validated the pipeline by sequencing artificial mixtures of *C. jejuni* strains and observed the highest prediction accuracy when including obtained isolates as references. By analyzing metagenomic samples, we were able to detect over 20% additional genetic diversity and observed an over 50% increase in the potential to connect genotypes across wild-bird samples. With an optimized filtration method and a computational approach for genotyping strain mixtures, we provide the foundation for future studies assessing *C. jejuni* diversity in environmental and clinical settings at improved throughput and resolution.

## 1. Introduction

*Campylobacter* infections are recurrently confirmed as one of the world’s leading causes of bacterial gastroenteritis. Luxembourg presents one of the highest incidences of *Campylobacter* infections in the European Union (EU), reaching an incidence rate of 116.4 cases per 100,000 population in 2020 [1]. Besides the major poultry-related transmission route, numerous other domestic animal and wildlife sources were identified [2]. The number and diversity of *Campylobacter* transiting in animal and environmental compartments is immense [3,4]. Wild birds are hypothesized to act as wildlife reservoirs of *Campylobacter jejuni*, causing fecal contamination of the environments, of food production farms, and possibly transmitting *Campylobacter* to livestock, domestic animals as well as to humans [5]. Aside from being confirmed in more than 80% of campylobacteriosis cases, *C. jejuni* is also the most frequently detected species in several wild bird species [6,7]. To investigate its origin, transmission routes and impact, and thus the risk of exposure, *C. jejuni* needs to be intensively monitored in clinical, environmental, and animal settings and efficient detection, isolation, and characterization tools must be developed.

Most of the standard methods for detection of *Campylobacter* recommend incubation in selective enrichment broth [8,9,10]. However, enrichment cultures have some drawbacks: (i) the overgrowth of unwanted bacteria or missing of part of the *Campylobacter* populations; (ii) competition and fitness-based selection of part of the *Campylobacter* community to the detriment of others [11,12]. All of this leads to the fact that the current picture of *Campylobacter* diversity in animal hosts and in the environment is probably incomplete. To avoid potential biases associated with enrichment culturing, passive-filtration methods (PFM) [13] can be applied that take advantage of the small shape and elevated motility of *Campylobacter* cells, inciting their passage through pores of filtration membranes while other bacteria are retained. For the identification of *Campylobacter* strains, microscopy and biochemical tests are the most common methods with confirmation by (multiplex) PCR [10]. Current molecular typing technologies, such as Multi-Locus Sequence Typing (MLST), Comparative Genomic Fingerprinting (CGF), or more recently, whole-genome sequencing, permit us to attribute groups of subtypes more and more tightly to specific hosts [14,15,16]. Here, we applied a typing scheme that improves the resolution over MLST by adding two additional loci, *porA* and *gyrA*, while maintaining comparability to previous studies utilizing sequence types (ST) [17]. Additionally, this extended-MLST scheme of 9 loci (hereinafter referred to as 9MLST) provides high concordance with cgMLST schemes [18].

Metagenomic sequencing can alleviate some of the issues inherent to isolate culturing approaches and to culture-independent diagnostic tests, however resolving complex mixtures of highly similar bacterial strains is still an area of active research. Recently, several approaches have been suggested to characterize strain diversity in metagenomic data [19,20] or capture strain-specific haplotypes by utilizing long sequencing reads [21,22]. Genotyping of strains from metagenomic data is frequently limited to the majority genotype [23] or requires extensive computational resources [24]. Particularly for *Campylobacter*, resolving strain mixtures is a complex task due to frequent horizontal gene transfer events [25] and high genetic microdiversity [26]. To achieve sufficient sequencing depth for strain deconvolution, plate sweeps offer the possibility to only target the organisms of interest and reduce potential sources for bias [27].

In this study, we assessed *C. jejuni* diversity by isolation and metagenomic analyses of surface water and wild bird fecal samples after PFM. To this end, we developed a computational workflow for improved strain-typing from metagenomic resolving 9MLST profiles and quantified the additional *C. jejuni* diversity captured within the wild bird reservoir, as well as its potential for reconstructing putative transmission routes.

## 2. Materials and Methods

### 2.1. Wild Bird Feces and Surface Water Sampling

In total, 362 fecal samples were collected from 43 wild bird species, in collaboration with the Luxembourg bird protection association Natur&Emwelt, between February 2019 and June 2021. Fecal samples were collected in 10 mL of sterile phosphate-buffered saline (PBS) and stored on ice for less than 6 h. During the same period, 87 surface water samples were collected from the Alzette, Moselle (including Remerschen ponds), Sûre rivers and the Weiswampach lake.

### 2.2. Passive Filtration Method (PFM)

Fecal samples were well-vortexed and 400 µL were transferred on a 0.65 µm filter (Sartorius cellulose nitrate membrane filter, diameter 47 mm) placed on modified-charcoal-cefoperazone-deoxycholate agar (mCCDA) plates (Oxoid, PO5091A) (Figure 1). The filter was removed from the agar surface after 18–20 h of incubation at 42 °C in sealed jars (Anaerojar 2.5L, Oxoid) under microaerobic conditions (CampyGen gas-generating system, Oxoid, CN0025). Between 24 and 72 h after supplementary incubation under microaerobic conditions, *Campylobacter*-like colonies were selected (up to ten isolates per sample) and isolated by multiple streaking steps on mCCDA and chocolate agar with Vitox (Oxoid, PO5090A). The remaining biomass on mCCDA plates after filtration were collected with a swab for metagenomic analyses (Figure 1). Depending on the water turbidity, adapted volumes of 100 to 500 mL were filtered on a 0.65 µm filter and the filter was placed retentate-up on mCCDA plate and incubated as previously described.

### 2.3. Identification and Sequencing of Campylobacter Isolates

*Campylobacter* isolates were identified by *hipO/glyA* gene PCR amplification [28] or MALDI-TOF [29]. All isolates were stored at −80 °C in FBP medium [30].

DNA was extracted by using the QIAamp DNA Mini Kit (Qiagen, The Netherlands) according to the manufacturer instructions. DNA was quantified with the Qubit 2.0 Fluorometer (Invitrogen, Belgium) and the Qubit^®^ dsDNA HS Assay kit (Life Technologies, Belgium). The DNA concentration was adjusted to be within the range of 3 and 17 ng/µL for subsequent sequencing. Libraries were prepared using the Nextera^TM^ DNA Flex Library Prep Kit (Illumina, USA), fragment size was assessed with the Bioanalyzer 2100 (Agilent, USA), and libraries were sequenced on the Miseq platform (Illumina, USA) yielding 250-bp paired-end reads. The paired-end raw read data were de novo assembled using Spades v.3.11.1 (default parameters) implemented on Ridom SeqSphere+ v8.3.1 (Ridom GmbH, Germany) [31]. Allelic profiles of MLST loci, *gyrA,* and *porA* were determined by comparison with the PubMLST loci database (see below).

### 2.4. Sequencing of Filter Sweeps and Validation Samples

Artificially mixed samples were prepared by selecting a panel of strains including 17 genetic profiles isolated from wild birds and 7 profiles regularly isolated from human cases (Appendix A). Before mixing the samples, the extracted DNA of these strains was adjusted to the same concentration of 17 ng/µL by using the elution buffer included in QIAamp DNA Mini Kit (A.E. buffer, Qiagen, The Netherlands). A total of 20 artificial mixes (M1 to M20) were completed with different ratios between the strains. The total number of strains was either 2 (n = 6 mixes), 3 (n = 11 mixes), or 4 (n = 3 mixes). Some mixes used the same set of data but with different ratios of the strains: 1 set of 3 strains for M1, M9 and M10; 1 set of 2 strains for M11, M12 and M13; 1 set of 3 strains for M15 and M16. An additional artificial mix was prepared by mixing 3 strains into a metagenome extract originated from poultry feces (processed by using the DNeasy PowerSoil kit, Qiagen, The Netherlands). This mix, referred to as “Spiked_Mix_P1_2” was artificially contaminated with an endemic strain identified in Luxembourg (i.e., lineage A, strain Camp034 [18]) and 2 other strains isolated from surface water in 2011 and 2012, respectively (see Appendix A). The final ratio of DNA in this mix was as follows: 11.4% lineage A, 5.7% of strain E110047, 2.8% of strain E120126 and 80% metagenome extract, yielding theoretical proportions of 0.57, 0.29, and 0.14 for the three strains.

All filter sweeps samples and artificial mixes (referred to as validation sample set) were sequenced on the Nextseq 1000 (read length 150) or Miseq (read length 300) Illumina platforms. Library preparation was carried out by using the NexteraTM DNA Flex Library Prep Kit (Illumina, USA). Raw read quality was assessed with FastQC v0.11.9 [32] and summarized with multiQC [33].

### 2.5. Genotyping of Metagenomic Samples from Filter Sweeps

A computational pipeline for analysis of metagenomic sequencing data was implemented with snakemake [34]. The pipeline contains individual workflows for preprocessing, running KMA [35] mapping, StrainEST [36] and consolidating the output for several samples, as well as auxiliary scripts for initial database construction and statistical analysis. Code is available under the following link: https://git.list.lu/malte.herold/campylomics_pipeline (accessed on 1 December 2022).

#### 2.5.1. Preprocessing

Adapter trimming and quality filtering of raw Illumina reads was performed with Trimmomatic v0.39 [37] using “NexteraPE-PE.fa” as adapter file and the following additional parameters (seed_mismatch: 2, palindrome_clip: 30, simple_clip: 10, leading: 20, trailing: 20, window_size: 1, window_quality: 3, minlen: 40, target_length: 40, strictness: 0.5). The remaining paired reads were retained for mapping.

#### 2.5.2. K-Mer Alignment (KMA) Step

Indices for KMA were generated for each of the 9MLST loci by first downloading the allele sequences from PubMLST REST API [38]. At the time of database generation (04/10/2022) the following numbers of sequences were downloaded for each of the respective loci: *aspA* (592 sequences), *gltA* (725), *glnA* (859), *glyA* (960), *tkt* (980), *pgm* (1286), *uncA* (757), *fn_gyrA* (412), *porA* (4600). After indexing, alignments against each of the 9MLST locus databases were performed for each sample by running KMA v1.3.13 for the paired end reads. Resulting hits were filtered according to the following criteria, determined from an initial analysis of the validation data set (synthetic mixtures). Query Identity (QI) was required to be at least 99.5%, while the cutoff for Template Identity (TI) was 50%, and the Q-value was required to be above 5. At higher sequencing depths, the requirement for QI was increased to 100% and the cutoff for TI was increased to 60% (Depth 5–10) or 70% (Depth above 10), as typically, low QI and TI at high depth was a sign of false positive hits.

#### 2.5.3. StrainEST Step

Three sources of reference sequences were utilized for the preparation of the strainEST database: PubMLST isolate genomes, genomes utilized in generation of the innuendo wgMLST typing scheme [39], and isolates genomes assembled in this study (Appendix A). Genomes were reannotated to obtain 9MLST profiles with the mlst tool (https://github.com/tseemann/mlst, accessed on 30 December 2022) using a custom scheme generated from the sequences downloaded for the KMA step.

We prepared three different StrainEST databases, the first one without isolates generated in this study (“NI” database) and less stringent filtering by sample location (any continent but Oceania and South America), the second one including isolates from this study (“WI”), and a third one also including isolates, but only including genomes from Europe (“EU”). The utilized databases are available under the following archive: https://doi.org/10.5281/zenodo.7377966 (accessed on 30 December 2022).

Inhouse isolates (for WI and EU) were selected by distinct 9MLST profiles and choosing the genomes with highest percentage of cgMLST hits and highest average coverage. The 9MLST allele specifications and provenance metadata were downloaded for 53,843 PubMLST genomes (28 October 2022) and filtered according to species “*C. jejuni*” and genome length between 1.5 and 1.85 Mbp, while also requiring 90% of assigned cgMLST (1343 loci) [40]. Reference genomes with distinct 9MLST profiles were selected with potentially novel MLST alleles allowed, while partial or missing alleles, as well as profiles with distinct multiple alleles for one locus were discarded. For genomes with identical 9MLST profiles preference for selection was given to genomes derived from “wild bird” samples, followed by the lowest number of contigs, the genome origin as Europe, and the most recent sampling date or most recent submission date. Genomes from Innuendo were selected by excluding genomes with origin USA (for EU database), and then selecting complete 9MLST profiles including novel alleles.

Genomes for database construction were compared by sourmash distances [41], with k-mer length 31, for outlier exclusion and to determine 10 mapping references by pam clustering [42] (with the R-packages, cluster and factoextra) through selection of medoids (Appendix A). Finally, the reference genomes for each of the respective databases were mapped against the type-strain genome (NC_002163.1) with StrainEST v1.2.4 (procedure described in [36]) (mapgenomes), SNVs were determined (map2snp), as well as their distance (snpdist), which was then clustered (snpclust), resulting in a final set of SNV clusters. Mapping references were aligned to the type-strain genome (mapgenomes) and a minimap2 [43] index was generated. Preprocessed reads for each sample were aligned with minimap2 v2.24-r1122 (preset sr) against the StrainEST mapping index followed by StrainEST profile estimation.

#### 2.5.4. Combining KMA and StrainEST Outputs

The 9MLST profiles of all members for each SNV cluster were scored according to whether alleles were detected with KMA in the same sample. The resulting score was adjusted for SNV clusters with lower abundance by not scoring identical allele matches already present in higher abundance clusters. This step was added to avoid false positive predictions by StrainEST inferring similar profiles at lower abundances. The SNV cluster member with the highest scoring profile according to KMA matches was then selected with a minimum adjusted score of one allelic match in KMA. Ties among SNV cluster members with highest scoring profiles were resolved by selecting those with the highest mean q-value to depth ratio. Additionally, only StrainEST predictions with abundance greater than 0.01 and max_ident over 0.99 were considered in the filtered output. Similarly as for the initial KMA thresholds, cutoffs were initially determined by evaluation of the results of the benchmarking dataset.

### 2.6. Statistical Analyses and Comparison of Known Mixtures—Recovered Isolates

Filtered output files from the KMA and StrainEST step was processed with a custom R script (compare_strainest_filt.R). In the beginning, individual sequencing replicates (for identical samples sequenced several times) were selected by highest mean StrainEST R^2^ value and highest MinDepth across the three runs for each of the databases (Appendix A).

In general, statistical analysis and processing of tabular data was performed with R 4.0.3 [44] with frequent utilization of the tidyverse [45] package. The ggstatsplot [46], the ggraph [47], and the igraph [48] packages were also utilized.

For the comparison of known mixtures, the accuracy of the profiles predicted from metagenomic data were assessed. Correctly matching profiles (at least 7 of 9 allelic matches) were counted as true positives, while false negatives were defined as profiles from the truth set not present in the set of predicted profiles. Predicted profiles not matched to the profiles of known strains were considered false positives. For each sample, recall was calculated as number of true positives divided by the sum of true positives and false negatives, while precision was calculated as number of true positives divided by the sum of true positives and false positives. The F1 score represents the harmonic mean of precision and recall. For matching profiles, the absolute error was calculated as the difference between known proportions in the truth dataset and predicted abundances.

For the comparison of recovered isolates and metagenomic samples, two pairs of samples (B025 and B027, B189 and B198) were excluded from the comparison as these were likely mislabeled during processing, here exact profile matches to a recovered isolate genome could be found in the respective other sample. Samples for which isolate and metagenomics data were available were compared and screened for exact matches (assigned reference genome corresponds to an isolate sequence included in reference database construction, WI and EU), 9MLST matches (the full allelic profile of the isolate matches a predicted profile), or partial matches (at least 7 of 9 loci are allelic matches). For a combined set of metagenomic and isolate profiles, all isolate profiles were utilized, and matching metagenomic profiles (any type of match) were omitted. Metagenomic profiles without a matching isolate (marked as “additional strain”) were then added to the combined set (Appendix A) utilized for the assessment of diversity and connectivity of the samples. Here, predicted profiles were filtered more stringently (corrected score above 3) to avoid the inclusion of false positive predictions and to provide a conservative estimate of additional diversity.

## 3. Results

### 3.1. Overview of Isolated Strains

Among the mCCDA plates with biomass growing after filtration, 87 and 14 filter sweeps were collected for metagenomic analyses from wild bird fecal and surface water samples, respectively (Table 1). From this collection of samples, 94 distinct *C. jejuni* isolates were obtained from 83 samples including 90 from wild bird fecal samples and 4 from surface water (Appendix A). Overall, isolates represent diverse genotypes with 59 distinct sequence types (ST), with ST45 and ST383 being the most frequently assigned with 6 strains each. The majority of samples yielded only one distinct isolate whereas two or three distinct strains were isolated from seven and two bird samples, respectively. Additionally, 25 *C. jejuni* were isolated from wild bird feces (n = 20) and surface water (n = 5) samples for which no metagenomic data was generated. These additional isolates were also included for database reconstruction for two of the databases (WI and EU). Additionally, genetic material from isolates was utilized in the preparation of synthetic mixtures.

### 3.2. Overview of Metagenomic Sequencing Data

*C. jejuni* was identified in 115 sequenced unique metagenomic samples, consisting of 85 bird samples, 14 water samples, and 21 artificial mixed samples for validation. For the wild bird samples prioritized in this analysis, on average 12.0 million raw reads were obtained by Nextseq sequencing (n = 85, sd = 6.2 × 10^6^) and 11.8 million reads (n = 85, sd = 5.9 × 10^6^) were retained after pre-processing (Appendix A). Samples (re)sequenced with Miseq showed a lower average raw read count (1.4 × 10^6^) and only 1 out of 14 Miseq libraries was selected for downstream analysis. However, due to the increased read length, similar R^2^ values were achieved in StrainEST (Appendix A).

For the StrainEST databases a total of 7041, 7006, and 3318 reference genomes were selected, respectively for NI, WI, and EU and mapped to the *C. jejuni* type-strain to determine the following number of SNVs: 29,408, 29,742, 86,719 and SNV clusters: 1035, 1009, 751. The increased number of genomes in the NI and WI databases reduces the overall number of SNVs as these are only called for regions aligned across all reference genomes.

### 3.3. Prediction of C. jejuni Profiles from Artificial Mixed Samples

For testing the reference-based workflow for predicting 9MLST profiles from metagenomic data, 21 samples consisting of known strain mixtures of two to four strains were analyzed (Appendix A) with a total of 60 strains across different samples including 25 unique strains. Overall, 405 (NI database, 84%), 465 (WI, 97%), and 471 (EU, 98%) of 480 distinct 9MLST alleles per sample were recovered correctly, i.e., were contained in the predicted strain profiles of the corresponding samples, while 117, 44, and 41 distinct predicted alleles were not contained in the truth set (false positives) (Figure 2A).

A comparison of complete profiles was performed based on matching profiles with at least 7 alleles of the 9MLST loci matching a strain from the truth dataset in the same sample. The number of strains was predicted correctly in 13, 15, and 16 out of the 21 samples for the three databases NI, WI, EU, respectively. Even though two allelic mismatches were allowed, the number of correctly recovered strains strongly corresponded to whether complete profiles were present with a reference genome in the StrainEST database. For the NI database only 11 of the 25 unique profiles, utilized in the synthetic mixtures, were present, while for the other two databases 21 profiles were contained. With these two databases (WI and EU), recovery of the correct synthetic mixtures was possible at high precision and recall (Figure 2B) reflected by F1-scores of 0.95 (WI) and 0.96 (EU), while for the NI database the F1-score was calculated at 0.75.

For matching strain profiles, we compared predicted abundances from StrainEST to known proportions utilized for the mixtures and observed a mean absolute error between 0.07 and 0.10 (Figure 2C), with abundance ranks matching in 60% (WI and EU) and 33% (NI) of cases.

Inspecting individual matches, we saw that differences in predicted 9MLST profiles occurred primarily for low abundance strains or when strains with the exact 9MLST profile were not available as references, while false positive predictions seemed to occur when very similar strains were present in the database that the method could not distinguish (Appendix A). For three mixtures, M11, M12, and M13, reference strain B038-110619-02, ST-*gyrA*-*porA* profile 10811-36-1279, was predicted consistently to be contained in the mixture, which likely is related to a contamination of isolate B038-110619-01 that was utilized to prepare the synthetic mixtures. Running the metagenomic workflow for the sequencing reads used to assemble the isolate genome B038-110619-01, the profile of B038-110619-02 was also detected at 8% abundance (Appendix A). For two other false positive predictions for the EU database a similar pattern could be observed: the profile 6427-11-1157 was predicted in samples M4 and M20, both of which contain reference strain B037-110619-02. However, from the sequencing reads of this isolate the profile 6427-11-1157 is inferred at only 2% abundance and max_ident 0.986 (Appendix A). It is possible that these predictions are not false positives, but instead reflect correctly recovered impurities in the isolated strains.

### 3.4. Comparison of Recovered Isolates and Predicted Profiles

In order to further assess the efficiency of the workflow for *C. jejuni* strain deconvolution, we compared the predicted 9MLST profiles to those recovered from sequencing individual cultured isolates. Metagenomic data and recovered isolates were available for a total of 83 samples, including 79 wild bird samples, of which 75 were used in the comparison (see methods). As isolates were utilized in the generation of two of the three StrainEST reference databases (see methods), it was also possible to quantify exact matches. A total of 66 inhouse isolate genomes were utilized in the reference databases (“WI” and “EU”), of which 54 originated from samples with metagenomic data and 48 (WI), respectively 49 (EU) were recovered as exact matches in the analysis (Figure 3).

Overall, of 828 alleles (9 loci for 92 isolate strains), 479 (58%), 764 (92%), and 774 (93%) were retrieved in the same samples for the three databases NI, WI, and EU, respectively, with a total of 175, 190, and 176 predicted strain profiles for each of the databases. A total of 90 distinct isolates were assessed and 36 (NI), 79 (WI), and 80 (EU) isolate strains were recovered with full 9MLST profile and 56, 86, 87 isolates were recovered with at least seven allelic matches (Figure 3 and Appendix A). The isolates not recovered likely constitute strains not present in the reference databases or contained at too low abundances within the metagenomic samples, e.g., for database EU performing best overall, three strains were not detected: B038-110619-03, B100-300719-02, W025-02122019-02.

Strain profiles predicted at higher abundance in StrainEST showed, in general, the tendency to match with an isolate from the same sample (Appendix A). This is a consequence of high abundance strains being more likely to be picked for cultivation and individual strains (not contained in mixtures) are reliably predicted at high abundances. In total, 48 of the samples utilized in the comparison did not contain a detectable mixture of strains in any of the three different runs. Interestingly, when not including isolates as reference genomes for the StrainEST database (NI), several of those individual genotypes were not detected, as no reference genome with sufficient similarity was included. However, these genotypes would have easily been detected solely by mapping to reference alleles with KMA. Some of the predicted profiles, particularly for database NI, are likely to be false positives with several predictions very similar to recovered isolates.

Overall, only a few samples appeared to be obviously problematic for the analysis workflow. Potentially, mixtures with several similar strains and thus overlapping 9MLST alleles are challenging for the KMA scoring algorithm, as well as recovering low abundances strains due to insufficient coverage. Examples of such potentially problematic samples include B035, B036, B037, where several closely related strains are predicted some of which might constitute false positive predictions. In general, also the surface water samples were challenging and, e.g., the isolate profile from W025 was not recovered in the metagenomic analysis.

### 3.5. Assessment of Additional Diversity Detected by Metagenomic Sequencing

For the assessment of additional recovered diversity, the results of the analysis based on the EU database were considered and to reduce the possibility of including false positive predictions, only profiles with a corrected score above three were considered. According to these criteria, metagenomic sequencing and data analysis allowed recovering 43 additional strain profiles from samples for which also at least one *C. jejuni* isolate had been recovered, as well as 24 additional strain profiles from 17 samples for which no isolates could successfully be recovered (Appendix A). These 17 samples consisted of 7 wild bird samples (B035, B076, B077, B078, B080, B082, B155) and 10 surface water samples (W022, W026, W053, W054, W078, W079, W081, W082, W083, W084). Combining additional predicted and filtered profiles and profiles from isolate genomes resulted in a combined set of 181 profiles derived from 123 samples (Appendix A). The most prevalent STs captured throughout the sampling campaign were 383, 45, and 677 (Table 2) and in total 76 distinct STs, respectively 86 9MLST profiles, were recovered in the combined dataset. Through metagenomic sequencing, 17 distinct STs, i.e., 22 distinct additional 9MLST profiles, were added, so contributing to over 20% of additional diversity.

However, one of the major advantages of metagenomic data allowing for the detection of strain mixtures is the increased potential to discover links between samples and thus hosts or reservoirs of *C. jejuni*. Connectivity by linking individual samples by shared 9MLST profiles is greatly improved when adding the profiles inferred from metagenomics (Figure 4). There are 55 edges for the dataset based on isolates (44 nodes with at least 1 shared profile) from 94 genotypes and 140 edges for the combined dataset (with 156 genotypes from 100 samples) with an average degree value of 2.50 and 3.89, respectively.

Overall, in the dataset including strains derived from metagenomics, migratory birds showed a lower level of connectivity, with mean degree 2.86 (n = 21) compared to sedentary birds, mean degree 3.47 (n = 58). In particular, corvids showed a high prevalence of *C. jejuni* (76 strains with distinct 48 9MLST profiles detected in 42 samples) as was demonstrated previously [49]. Additionally, several links to surface water samples could be inferred by the additional metagenomics analysis that were not detected in the recovered isolates.

## 4. Discussion

In our comparison of isolate sequencing and strain-typing with filter sweep metagenomics with a reference-based approach, we show that metagenomics allows for increased detection and resolution of *C. jejuni* strain diversity and is particularly advantageous also for the recovery of lowly abundant strains from mixtures. However, the detection is limited by sequencing depth and with decreasing depth, the detection of lowly abundant strains becomes less and less reliable. A challenge in method development was the tendency of StrainEST to predict low abundance false positives closely related to strains present in the mixture, which was already highlighted by the authors in the original publication [36]. Here, we utilized a mapping-based approach to score StrainEST output and improve overall predictions as well as for selecting the most accurate cluster representatives, increasing the resolution of the method. While with this method we were able to resolve mixtures on the level of 9MLST profiles, a similar procedure could also be applied on the level of cgMLST or wgMLST schemes [40] which could be even more capable to distinguish mixtures of highly similar strains albeit requiring more computational resources.

For reference-based approaches, a major limitation is the reconstruction of suitable databases, i.e., including enough reference genomes to detect environmental *C. jejuni* strains, but too many reference genomes lead to a reduced number of SNVs when requiring SNV calling only for genomic regions aligned across all references as in StrainEST. Here, a trade-off in the selection of reference genomes had to be realized by selecting genomes with distinct 9MLST profiles and with high contiguity. Additionally, sample location and other provenance data were used as features to reduce the number of selected reference genomes. In this study, the reference database without sequenced isolates obtained from the same samples (NI) performed much worse in the benchmarking comparison and in the recovery of isolate-specific profiles than the two databases including those isolates (WI, EU). This is a consequence of the genotypes recovered by isolation or similar strains not being sufficiently represented in the reference databases and indeed several of the isolates recovered in this study represent novel genotypes. Of the two databases with isolates, restricting the location of the reference genomes to Europe slightly improved performance, potentially by reducing the total number of genomes, leading to higher coverage and more SNV-calls improving the accuracy of the StrainEST step. To tailor the reference database to a broader resolution, refined clustering approaches suitable for large-scale genome collections [50] could also be applied. Overall, also long-read sequencing methods have enormous potential to resolve strain mixtures either by correction of assemblies with long-read mapping [21] or direct haplotype-aware assembly [51], with the advantage of enabling the recovery of novel strains.

Here, we showed that the developed method can reliably be used for genotyping of low-complexity mixtures, as well as samples with individual strains. We recovered additional diversity compared to isolate-cultured based efforts. Over 20% additional STs were recovered in a conservative estimate of around 67 additional strains recovered from the metagenomics data. Another advantage of metagenomics, highlighted in this study, is the improved connectivity, by the detection of similar strains across several samples. This is particularly useful for studies tracing specific genotypes across different environments and hosts. Producing a more complete picture of mixed genotypes in environmental samples can be relevant for outbreak investigation when groundwater and surface waters are contaminated by multiple sources during heavy rainfall [52]. This strain-typing metagenomics strategy could also be useful for epidemiologic surveillance in wastewater treatment plants, to provide an overview of circulating strains similarly to the recently implemented monitoring of SARS-CoV-2 variants over these last years [53].

The increased recovery of *C. jejuni* profiles by metagenomics highlights the benefits of the approach for epidemiological studies. Among the most prevalent STs recovered in wild birds (top-7, corresponding to 27% of detected profiles), we noted ST45 and ST19 which are well-known STs commonly isolated in Europe and are typically found in a broad range of environments [18,54,55]. Other commonly detected STs such as ST677 and ST383 have been previously observed in clinical cases and poultry samples [56,57] and could represent candidates to further explore potential routes of transmission. All the previously mentioned generalist lineages were detected in migratory as well as sedentary birds, while ST448, which is only sporadically found in clinical samples but frequently observed in wild birds [58], was solely detected in sedentary birds. In the set of prevalently observed STs, we also observe two novel profiles with ST11383 and ST11379 solely detected in sedentary birds, and which could be specific to local populations. Overall, STs discovered in this study were often just detected in one sample (61%) and could not be assigned to clonal complexes, highlighting the genomic diversity of *C. jejuni* populations carried by wild birds.

We are confident that the method introduced here could also be applied in a clinical setting, without the requirement of generating additional reference genomes by isolation as clinically relevant genotypes are more prevalent in public repositories. However, culture-independent diagnostic methods, including mass-spectrometry, immunological detection methods, flow cytometry, or PCR are used to an increasing extent, which could lead to a lack of suitable reference genomes in future studies. Furthermore, these methods do not provide subsequent information for subtype analysis, molecular epidemiology, or for the detection of antibiotic resistances [10].

Even though, the approach demonstrated here was developed for the detection of *C. jejuni*, similar approaches could be applied for discriminating low-complexity mixtures of distinct strains at high resolution for other species. Genotyping directly from metagenomics data could be implemented in clinical and environmental surveillance approaches, potentially with an additional enrichment step for low biomass samples.

## Figures and Tables

**Figure 1 microorganisms-11-00121-f001:**
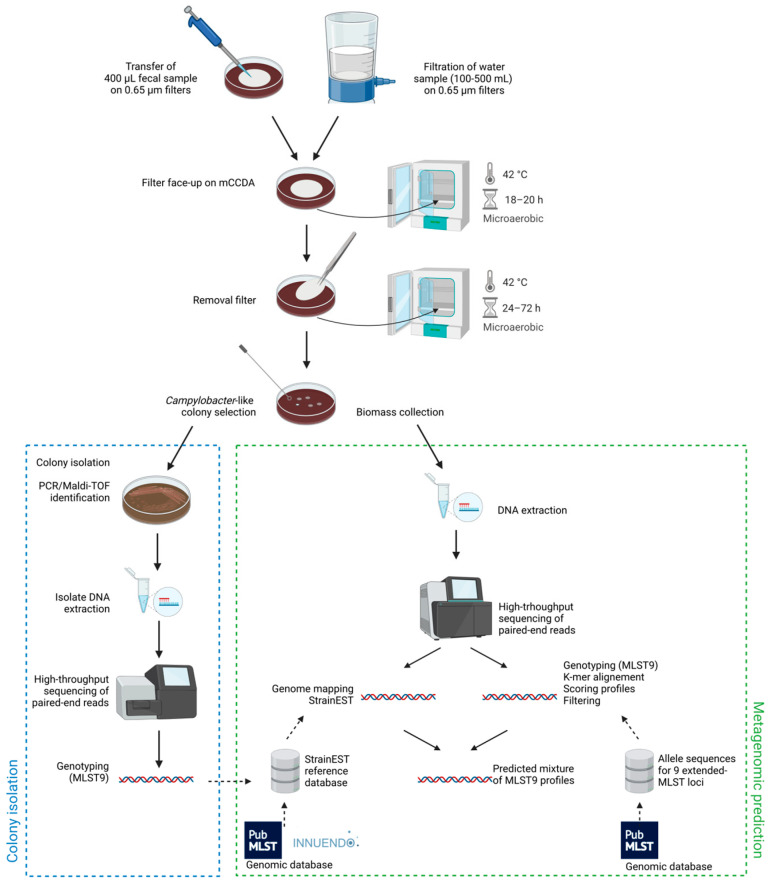
Overview of the filtration procedure, isolation, and metagenomic typing workflows. Created with BioRender.com.

**Figure 2 microorganisms-11-00121-f002:**
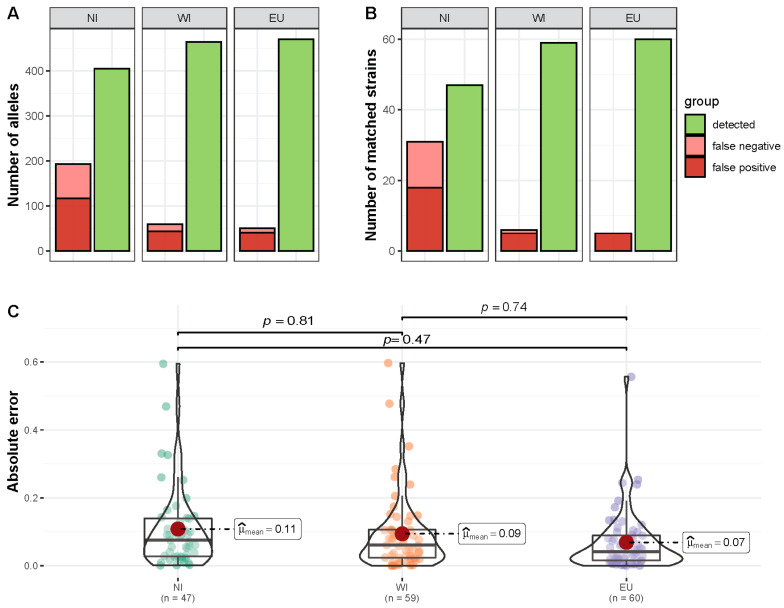
Summary of the prediction of *C. jejuni* strains from artificial mixed samples. (**A**) Overall numbers of correctly recovered 9MLST alleles per sample in the prediction set. False negatives are unique alleles present in the ground truth, but not detected in the prediction set, while false positives are predicted alleles not present in the strains utilized for synthetic mixtures. (**B**) Overview of matching inferred strain profiles and known mixtures per sample. A total of 21 known mixtures and predicted profiles were assigned as matches if 7 out of alleles of the 9MLST profile (ST-gyrA-porA) were correct. False negatives are strains from the truth set without a match, while false positives are inferred profiles not present in the truth set (**C**) Absolute differences between known proportions (as used for the synthetic mixtures) and predicted abundances (StrainEST) for matching strains for the three databases, NI (database without isolates from this study), WI (database including isolates from this study), and EU (database including isolates from this study and only reference genomes from Europe). *P*-values for pairwise comparisons (Games-Howell test) between groups are adjusted with the Holm method.

**Figure 3 microorganisms-11-00121-f003:**
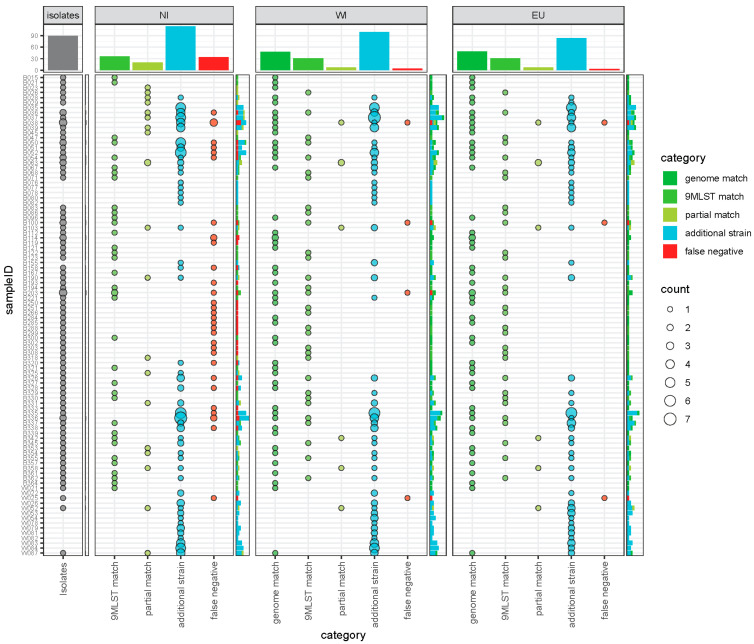
Overview of predicted 9MLST profiles compared to isolated strains according to categories of matches (Appendix A). Panels are separated according to the utilized database (NI: without isolates from this study, WI: including isolates from this study, EU: including isolates from this study and only reference genomes from Europe) with a separate panel highlighting the number of recovered isolates on the left. Sample identifiers starting with B represent wild bird samples and W stands for surface water samples. Point size indicates the number of strains assigned to each category.

**Figure 4 microorganisms-11-00121-f004:**
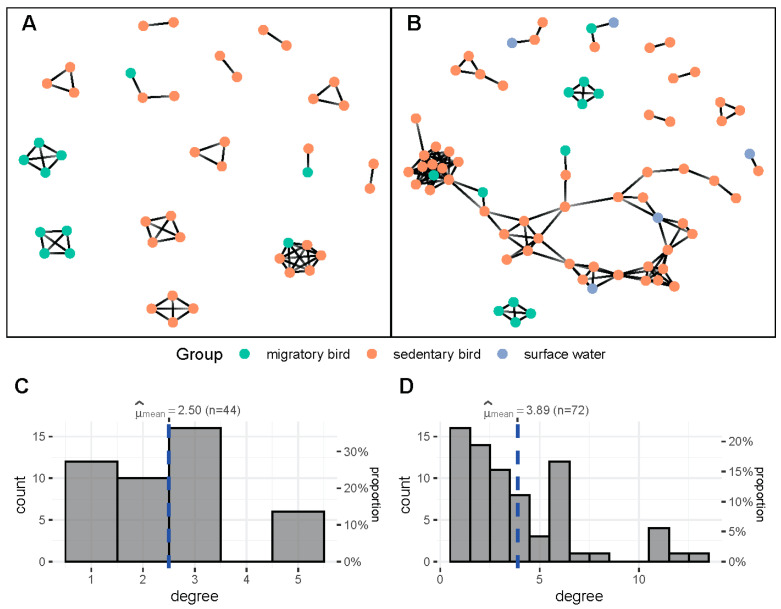
Network connectivity of individual samples based on co-occurrence of 9MLST profiles within the combined dataset (Appendix A). Each node represents an individual sample, colored according to its origin. (**A**) Network based on isolated strains. (**B**) Network based on the combined dataset of isolated strains and metagenomic samples. (**C**) Histogram of degree values for each node for the graph in (**A**). (**D**) Histogram of degree values for each node for the graph in (**B**).

**Table 1 microorganisms-11-00121-t001:** Overview of *C. jejuni* strains isolated or detected by metagenomic analyses from wild birds and surface water.

	Wild Bird	Surface Water
Number of samples analyzed	87	14
*C. jejuni* isolation positive samples	79	4
*C. jejuni* metagenomically predicted positive samples	86 (72 *)	13
Number of isolated *C. jejuni* strains	90	4
Number of predicted strains (Europe database)	150	26
Number of predicted strains (WI-database)	159	31
Number of predicted strains (NI-database)	143	32

* NI database.

**Table 2 microorganisms-11-00121-t002:** Prevalent STs (at least 5 occurrences) in the combined dataset (isolates and metagenomics).

ST	#Profiles	Unique 9MLST Profiles	Profiles from Isolates	Migratory Birds/Sedentary Birds	Earliest Sampling Date	Latest Sampling Date
383	12	1	6	3/9	2 July 2019	7 June 2021
45	9	4	6	1/5	7 May 2019	25 May 2021
677	7	1	4	2/5	7 May 2019	21 June 2021
11379	6	1	2	0/6	11 June 2019	7 June 2021
11383	5	1	1	0/5	5 May 2019	7 June 2021
19	5	3	4	1/3	18 June 2019	7 June 2021
448	5	3	4	0/5	7 May 2019	21 June 2021

## Data Availability

Additionally to the data provided as Appendix A with this publication, the following sources contain data generated within this study: Zenodo archive of databases and raw analysis output: https://doi.org/10.5281/zenodo.7377966 (accessed on 30 December 2022). Raw sequencing reads metagenomics (Bioproject PRJEB55463). Raw sequencing reads artificial strain mixtures (Bioproject PRJEB55470). Raw sequencing reads for isolate sequences (Bioproject PRJEB57730). Code used: https://git.list.lu/malte.herold/campylomics_pipeline (accessed on 1 December 2022).

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
