# Peer review of "Metagenomic Strain-Typing Combined with Isolate Sequencing Provides Increased Resolution of the Genetic Diversity of Campylobacter jejuni Carriage in Wild Birds"

_microorganisms, 2023, doi:10.3390/microorganisms11010121_

Round 1

Reviewer 1 Report

The research question is important. The food- and waterborne pathogen Campylobacter needs to be intensively monitored through a One Health approach. The manuscript is proposing a novel method to obtain Campylobacter jejuni isolates with a passive-filtration method (PFM) and to analyze them by a metagenomic approach. It is a quite interesting way to produce more data for epidemiological analysis of this pathogenic bacteria.

The manuscript presents a well-written introduction to the research subject. The need to find a solution to the selection bias of some Campylobacter jejuni strains in the traditional bacterial isolation procedure is highlighted. However, it is also quite lengthy, detailing some unnecessary information and missing the main focus in some parts (such as the detailed explanation of how to perform the MLST - lines 60-68). The main objective could also be summarized in a short final sentence without detailing the different bioinformatic procedures such as KMA, 9MLST, strainEST. I suggest that this detailed methodological information be presented only in the Methods section.

Materials and Methods are OK, including a well-prepared figure to understand the whole procedure. But it is necessary to review the number of topics (the PFM was not included in the sequential numbering) as well as review the entire organization. I suggest joining topics 2.2 and 2.3 under “Identification of Campylobacter isolates” (including all methods to identify isolates). Furthermore, the explanation of the computational pipeline is quite extensive and a shorter description would be most welcome.

The results are well presented, but also quite extensive. It would take some effort to summarize them. On the contrary, the discussion is very short. It is limited to comparing the entire procedure developed and not the main lineages/strains in an epidemiological context. I was expecting more discussion about the epidemiological relevance of the most frequent STs detected in the study. Therefore, I suggest including a few paragraphs to discuss the genetic diversity findings and their epidemiological significance, comparing with other scientific studies.  

Finally, it is necessary to review all manuscript to write correctly scientific names, as for example genes names (always in italic).

Author Response

We thank the reviewer for their willingness to review the manuscript and their thoughtful comments. In the following please find a point-by-point reply to the comments. Please note that the attached word file contains the manuscript with track-changes instead of a response letter.

1. However, it is also quite lengthy, detailing some unnecessary information and missing the main focus in some parts (such as the detailed explanation of how to perform the MLST - lines 60-68). 

We appreciate the reviewer’s feedback and have shortened and streamlined the introduction where possible.   

2. The main objective could also be summarized in a short final sentence without detailing the different bioinformatic procedures such as KMA, 9MLST, strainEST. I suggest that this detailed methodological information be presented only in the Methods section. 

We have clarified and shortened this summary of the main objectives. 

3. Materials and Methods are OK, including a well-prepared figure to understand the whole procedure.   But it is necessary to review the number of topics (the PFM was not included in the sequential numbering) as well as review the entire organization. I suggest joining topics 2.2 and 2.3 under “Identification of Campylobacter isolates” (including all methods to identify isolates).  

We thank the reviewer for spotting this oversight and the practical suggestion and have revised the manuscript accordingly. 

4. Furthermore, the explanation of the computational pipeline is quite extensive and a shorter description would be most welcome. 

We appreciate the remark, however we believe that the level of detail provided here is necessary to ensure reproducibility. Nonetheless, we have shortened and simplified the description of the computational pipeline where possible.  

5. The results are well presented, but also quite extensive. It would take some effort to summarize them. On the contrary, the discussion is very short. It is limited to comparing the entire procedure developed and not the main lineages/strains in an epidemiological context. I was expecting more discussion about the epidemiological relevance of the most frequent STs detected in the study. Therefore, I suggest including a few paragraphs to discuss the genetic diversity findings and their epidemiological significance, comparing with other scientific studies.   

In the result section we directly discuss individual aspects of the methodology to be able to focus on the broader picture in the discussion section. We agree with the reviewer that discussing the role of the most common STs recovered in the study was lacking and have added a paragraph contextualizing their epidemiological relevance in the discussion. However, we’d like to note that the methodological aspects are more central to this study compared to the epidemiological interpretation of the results.    

6. Finally, it is necessary to review all manuscript to write correctly scientific names, as for example genes names (always in italic). 

We have carefully re-checked italicization of scientific names throughout the manuscript.

Reviewer 2 Report

Reviewer # (Comments for the Author):

The manuscript (microorganisms-2101297) entitled "
Metagenomic strain-typing combined with isolate sequencing provides increased resolution of the genetic diversity of Campylobacter jejuni carriage in wild birds" applied a passive-filtration method to obtain isolates and in parallel recover genotypes from metagenomic sequencing data from associated filter sweeps. For genotyping mixed strains, this study was utilized a reference-based approach by combining existing computational tools in a semi-automated workflow to predict allelic profiles of 9 extended-MLST loci.

The results of this study showed that the developed method can reliably be used for genotyping of low-complexity mixtures, as well as samples with individual strains. Over 20% additional STs were recovered in a conservative estimate of around 67 additional strains recovered from the metagenomics data. Another advantage of metagenomics, highlighted in this study, is the improved connectivity, by the detection of similar strains across several samples. This is particularly useful for studies tracing specific genotypes across different environments and hosts.

Even though, the approach demonstrated here was developed for the detection of C. jejuni, similar approaches could be applied for discriminating low-complexity mixtures of distinct strains at high resolution for other species. Genotyping directly from metagenomics data could be implemented in clinical and environmental surveillance approaches, potentially with an additional enrichment step for low biomass samples.

The manuscript is well organized and written in an understandable manner and the results of this study well analyzed and demonstrated in different tables, figures and in supplementary files.

General comment:

1.                  There are some grammatical errors need to be addressed by the authors to improve the manuscript and please clarify the abbreviation used (e.g. Line 92: PBS).

2.                  The authors need to write the abstract according to the results of the manuscript briefly.

3.                  In methodology section, line 123, the author defined the concentration of DNA, what about the quality?

4.                  I suggest to merge the result and discussion section together.

Author Response

We thank the reviewer for their thorough and considerate comments on the study. In the following please find a point-by-point response to the comments and the revised manuscript with track-changes enabled in the attachment.

  1. There are some grammatical errors need to be addressed by the authors to improve the manuscript and please clarify the abbreviation used (e.g. Line 92: PBS). 

We appreciate the comment and re-checked the grammar carefully and clarified the use of abbreviations throughout the manuscript. 

2. The authors need to write the abstract according to the results of the manuscript briefly. 

We re-wrote the abstract and highlighted specific results.  

3. In methodology section, line 123, the author defined the concentration of DNA, what about the quality? 

We performed quality control on the prepared libraries and have added a half-sentence to reflect this in this section.  

4. I suggest to merge the result and discussion section together. 

While we agree with the reviewer that merging the results and discussion section would be a good option in this case, the author guidelines of the journal state that a discussion section is required and after correspondence with the editor we decided to keep a separate discussion section.  

Round 2

Reviewer 1 Report

I revised the first version of this manuscript, suggesting further revision. The authors implemented most of the suggestions and the manuscript was improved. The number of topics in Materials and Methods has been reduced, but the explanation of the computational pipeline is still extensive and I think further effort to shorten it would be welcome. The authors also added a discussion of the suggested epidemiological relevance, further highlighting the importance of the developed method.

I also have another additional suggestion. A very excessive number of sentences are written in the first person plural “we” (especially in the Abstract, Objective and Discussion). I counted five only in the Abstract, apart from the rest of the text. Therefore, I recommend rewriting most of these sentences using the passive voice (which would be much more suitable for a scientific text).

Author Response

We appreciate the reviewers additional revision of the manuscript. We agree that the description of the computational pipeline is somewhat detailed, however we believe that this level of detail is not extensive but required and further shortening this section would require omitting details necessary for reproducticibility and for assessing our results (e.g. criteria for database construction).

The reviewer's assessment that using the passive voice is much more suitable for scientific writing is under debate within the scientific community. We believe that utilizing the active voice enables a clearer and more readable writing style and thus used it preferentially throughout the manuscript. We changed one sentence in the abstract to passive voice to avoid the appearance of excessiveness.
